# SupCL-GSS: Supervised Contrastive Learning with Guided Sample Selection

## Abstract

We present Supervised Contrastive Learning with Guided Sample Selection (SupCL-GSS), that leverages data maps to construct "hard" positives and "hard" negatives for text classification on pre-trained language models. In our method, we first measure training dynamics to identify the *learning difficulty* of each training sample with respect to a model—whether samples are easy-to-learn or ambiguous. We then construct positive and negative sets for supervised contrastive learning that allow guided sample selection based on both *samples' learning difficulty* and *their class labels*. We empirically validate our proposed method on various NLP tasks including sentence-pair classification (e.g., natural language inference, paraphrase detection, commonsense reasoning) and single-sentence classification (e.g., sentiment analysis, opinion mining), both on in- and out-of-domain settings. Our method achieves better performance and yields lower expected calibration errors compared to competitive baselines.

## 1 Introduction

Contrastive learning is a variant of self-supervised learning which does not require any labeled data (Wu et al., 2018; Tian et al., 2020); instead, it aims at optimizing the representations by minimizing the distance between similar samples (i.e., positive pairs) and maximizing the distance between dissimilar samples (i.e., negative pairs), to produce high-quality representations. A major focus in **self-supervised** contrastive learning is on constructing more challenging negative sets, i.e., hard negatives Robinson et al. (2021); Kalantidis et al. (2020).

Concomitantly, Khosla et al. (2020) extended self-supervised contrastive learning to *supervised* contrastive learning which constructs positive and negative sets guided by *samples' class labels*. For example, all samples from the same class as the anchor are considered as positives whereas samples from different classes are considered as negatives. Gunel et al. (2020) showed that including a supervised contrastive learning term in the overall loss yields promising results on the GLUE benchmark Wang et al. (2018). Furthermore, Sedghamiz et al. (2021) improved Gunel et al. (2020)'s performance by leveraging built-in dropout masks of a pre-trained language model along with class labels. However, these supervised contrastive learning works construct negative pairs by simply selecting random samples that have different class labels. Thus, devising hard negatives for **supervised** contrastive learning remains under-explored—and in fact, we argue that not only hard negatives, but also hard positives are equally important.

In this paper, we aim to construct *hard positive* and *hard negative* sets for supervised contrastive learning to learn better representations for text classification tasks on pre-trained language models. For this, we first utilize data maps Swayamdipta et al. (2020) to categorize training samples according to their *learning difficulty* with respect to a model as easy-to-learn or ambiguous, and then construct positive and negative sets that allow guided sample selection based on both *samples' learning difficulty* and *their class labels*. Data maps Swayamdipta et al. (2020) leverage the mean and standard deviation of the gold label probabilities, predicted by a model for each training sample across training epochs (referred as confidence and variability, respectively); the samples that the model predicts correctly and consistently (high confidence, low variability) across epochs are identified as easy-to-learn samples, whereas those with high variability for which gold label probabilities fluctuate frequently during training are identified as ambiguous for the model and are those for which the model is uncertain about. To illustrate the intuition for using samples' learning diffi-

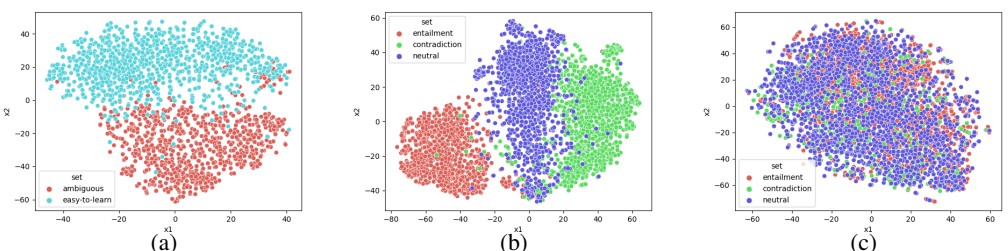

Figure 1: t-SNE visualization of 1,000 SNLI training samples as: (a) easy-to-learn and ambiguous from the *entailment* class, (b) easy-to-learn from all classes, and (c) ambiguous from all classes, with a BERT model.

culty in hard sets construction, we explore how samples align in the feature space along these two categories.

Hence, we show t-SNE visualization of easy-to-learn and ambiguous samples from the Stanford Natural Language Inference (SNLI) dataset Bowman et al. (2015) in Figure 1 as follows: (a) easy-to-learn and ambiguous samples that belong to the 'entailment' class; (b) easy-to-learn samples of all class labels (i.e., 'entailment', 'contradiction' and 'neutral'), and (c) ambiguous samples of all class labels. As shown in Figure 1a, we observe that samples that belong to the same class ('entailment') are likely to be separable into easy-to-learn and ambiguous samples. This is because the ambiguous samples are those that reside near the decision boundary and highly likely overlap with other classes whereas the easy-to-learn samples cluster together tightly further away from the decision boundary. In Figure 1b, we can see that easy-to-learn samples are fairly clustered along their classes. Last, we can observe from Figure 1c that ambiguous samples from different classes are not clustered along their classes and are those for which the model is uncertain about.

To this end, we propose **Sup**ervised **C**ontrastive **L**earning with **G**uided **S**ample **S**election (**SupCL-GSS**), for constructing hard positive and hard negative sets guided not only by *class labels* but also by *samples' learning difficulty* (i.e., as easy-to-learn or ambiguous). Specifically, to construct positive pairs for a given sample (anchor), we select samples that satisfy the following rules: (1) samples that have **the same class label** as the anchor and share **the same learning difficulty**, and (2) **the k most dissimilar** samples to the anchor that have **the same class label** as the anchor but **different learning difficulty**. For example, in Figure 1(a), easy-to-learn samples should stay close to each other since they share the same label 'entailment'. Our method pulls these samples together to encourage them to remain close in the feature space by satisfying rule (1). Furthermore, easy-to-learn and ambiguous samples that are most dissimilar but from the same class are likely to be located further apart in the feature space (see Figure 1(a)). Our method pulls these samples from the same class together to ensure they come close in the feature space by satisfying rule (2). Consequently, we encourage easy-to-learn and ambiguous samples to form a tighter cluster in the feature space. Furthermore, to construct negative pairs for a given sample (anchor), we select samples that satisfy the following rule: samples that have **different class labels** to the anchor but **the same learning difficulty**, and whose **cosine similarities** with the anchor are higher than a pre-defined threshold value. For example, in Figure 1(c), our method ensures that ambiguous samples that belong to the 'contradiction' and 'neutral' classes and that are similar to each other are pushed further apart. Thus, similar to Robinson et al. (2021), our method constructs the hard negatives to prefer examples that are (incorrectly) close to the anchor. However, when constructing both the hard positive and hard negative sets, we account for the certainty / uncertainty of the model in the samples' labels to provide additional signal.

Since we explicitly infuse the certainty / uncertainty of the model through samples' learning difficulties (as easy-to-learn or ambiguous), we comprehensively investigate the impacts of our method on both model performance and model calibration, i.e., the ability of the model to express its uncertainty in predicting labels for unseen test data, which is essential for model trustworthiness Guo et al. (2017).

Our contributions are as follows:

- We introduce supervised contrastive learning with guided sample selection called **SupCL-GSS** which constructs hard positive and negative sets guided by *samples' learning difficulty* and *class labels*, to enhance the generalization of representations learned for text classification on pre-trained language models.

- We demonstrate that SupCL-GSS not only achieves improved accuracy on both in-domain and out-of-domain settings compared to strong baselines on various NLP tasks, but also yields lower expected calibration errors.

- We analyze the contribution of each component of SupCL-GSS, and visualize the feature representations, showing the necessity of each component and the advantage of our method.

## 2 RELATED WORK

**Contrastive Learning**  Self-supervised contrastive learning has achieved remarkable success on many tasks and domains He et al. (2020); Chen et al. (2020); Kalantidis et al. (2020); Robinson et al. (2021); Gao et al. (2021); Wang et al. (2021); Chai et al. (2021); Liang et al. (2022); Qin et al. (2022); Park et al. (2023); Wan et al. (2023); Chen et al. (2024). Supervised contrastive learning overcomes the main shortcoming of self-supervised contrastive learning by considering the class labels of inputs Khosla et al. (2020); Gunel et al. (2020); Sedghamiz et al. (2021); Zeng et al. (2021); Guo et al. (2023). Many prior works on supervised contrastive learning focus on generating altered views for anchors to expand positive sets while simply generating negative views by leveraging class label information Sedghamiz et al. (2021); Zeng et al. (2021); Park et al. (2023). However, these works heavily rely on data augmentation to diversify positives, which may not effectively generate hard positives. Furthermore, they simply use class labels to construct negatives, which may not necessarily generate meaningful negatives. While Robinson et al. (2021) and Kalantidis et al. (2020) prove the benefits of hard negatives in self-supervised contrastive learning, existing research lacks methods specifically for constructing hard positives and hard negatives in **supervised** contrastive learning. We address this gap by leveraging both the samples' learning difficulty and class labels for constructing hard positives and hard negatives. We chose Kalantidis et al. (2020) and Robinson et al. (2021) as strong baselines for comparison due to their alignment with our work in targeting hard negatives (alongside other supervised contrastive learning baselines).

**Curriculum Learning**  Our method is reminiscent of curriculum learning Bengio et al. (2009); Wang et al. (2022); Nagatsuka et al. (2023), which mimics the human learning process (from simpler to more complex concepts). Curriculum learning designs a difficulty measure ahead, and an easy-to-difficult curriculum is arranged accordingly for the learning procedure. Nagatsuka et al. (2023) proposed to use the length of input text as the difficulty measure. Swayamdipta et al. (2020) ranked samples according to their learning difficulty and used data regions for model training. We use the methods by Nagatsuka et al. (2023) and Swayamdipta et al. (2020) as additional baselines in experiments.

**Model Calibration**  A well-calibrated model ensures that the model's confidence in its predictions reflects its actual accuracy Guo et al. (2017). Prior works show that applying either self- and fully-supervised contrastive loss term as a regularizer enhances model calibration on image classification Liu & Abbeel (2020); Tack et al. (2020); Winkens et al. (2020); Khosla et al. (2020), and on graph representation learning Ma et al. (2021); Zhang et al. (2023). However, there is a notable gap in research as no prior works delve into the implications of model calibration in supervised contrastive learning on text data using pre-trained language models. Most prior works on NLP focus on performance improvement using contrastive learning on various tasks such as out-of-domain intent detection in a task-oriented dialogue system Zeng et al. (2021), relation extraction Wan et al. (2023); Guo et al. (2023) and code search with question answering Park et al. (2023), rather than model calibration. In contrast, we show that SupCL-GSS not only achieves better performance but also yields lower expected calibration errors on both single sentence and sentence-pair classification tasks.

## 3 PROPOSED APPROACH

We now introduce Supervised Contrastive Learning with Guided Sample Selection named **SupCL-GSS**. We first describe the process of identifying the learning difficulty of each sample by leveraging training dynamics (i.e., confidence, variability) (§3.1). We then present our method of constructing positive and negative sets in SupCL-GSS, guided by both the samples' learning difficulty and class label (§3.2).

### 3.1 IDENTIFYING THE LEVEL OF DIFFICULTY

We describe **confidence** and **variability**, two statistics that are used to identify the level of difficulty of each sample Swayamdipta et al. (2020). To obtain these statistics, we fine-tune a pre-trained language model in advance to calculate them for each sample $(x_i, y_i)$ over $E$ training epochs.

**Confidence** is the mean of the model probability of the true (gold) label $y_i$ across epochs:

$$\hat{\mu}_i = \frac{1}{E} \sum_{e=1}^{E} p_{\theta^{(e)}}(y_i | x_i)$$

where $p_{\theta^{(e)}}$ denotes the model's probability with parameter $\theta^{(e)}$ at the end of the $e^{th}$ epoch.

**Variability** is the standard deviation of $p_{\theta^{(e)}}$ across epochs $E$:

$$\hat{\sigma}_i = \sqrt{\frac{\sum_{e=1}^{E} (p_{\theta^{(e)}}(y_i | x_i) - \hat{\mu}_i)^2}{E}}$$

Given these statistics per sample, we identify the *learning difficulty* of each sample. If a model predicts a sample correctly and consistently across epochs (high-confidence, low-variability), we identify it as an easy-to-learn sample. Otherwise, if a sample whose true class probabilities have a high variance during training (high-variability), we identify the sample as ambiguous. To leverage the most representative easy-to-learn and ambiguous samples from a training set, we rank samples based on confidence and variability, respectively, and select the top-ranked training samples. In experiments, we rank all training samples by confidence in descending order and select the top 33% samples to construct the easy-to-learn set $\mathcal{D}_{easy}$. Similarly, we rank all training examples by variability in descending order and select the top 33% samples to obtain the ambiguous set $\mathcal{D}_{ambig}$. Our choice of top 33% easy and 33% ambiguous samples was inspired from Swayamdipta et al. (2020), but we explored other ratios (e.g., 25%, 50%) in experiments.

### 3.2 SUPERVISED CONTRASTIVE LEARNING WITH GUIDED SAMPLE SELECTION

We propose supervised contrastive learning that is guided by both the samples' learning difficulty and class labels. For a sample $x_i$ we obtain its sequence embedding $\mathbf{x}_i$ (i.e., feature representation) from the last layer of a pre-trained language model $f$.

**Constructing Positive Sets** We observe from Figure 1(a) that even when samples belong to the same class ('entailment' in the figure), they still form clusters along easy-to-learn and ambiguous sample sets. Thus, for each sample (anchor) that belongs to class $c$, we build its positive set with: (1) all samples that belong to $c$ and share the same learning difficulty as the anchor; and (2) the $k$ most dissimilar samples that belong to $c$ but have opposite learning difficulty. For example, if an anchor is easy-to-learn $(\mathbf{x}_i, y_i) \in \mathcal{D}_{easy}$ and has class label $c$ (i.e., $y_i = c$), we then construct the positive set $P(i)$ by the following two types of samples: (1) all easy-to-learn samples $(\mathbf{x}'_i, y'_i) \in \mathcal{D}_{easy}$ that belong to $c$, and (2) $k$ ambiguous samples $(\mathbf{x}''_i, y''_i) \in \mathcal{D}_{ambig}$ whose cosine similarities on hidden representations are the $k$ smallest compared with $\mathbf{x}_i$, and have the class label $c$. Formally, our positive set construction $P(i)$ is as follows:

$$P(i) = \{(\mathbf{x}'_i, y'_i) \in \mathcal{D}_{SameDiffic} : y'_i = c\} \cup \{(\mathbf{x}''_i, y''_i) \in \mathcal{D}_{Opp'lDiffic} : y''_i = c, \arg\min\langle \mathbf{x}''_i, \mathbf{x}_i \rangle\}$$
(1)

where $\mathcal{D}_{SameDiffic}$ denotes the same learning difficulty as the anchor, $\mathcal{D}_{Opp'lDiffic}$ denotes the opposite learning difficulty to the anchor, and $\langle \cdot, \cdot \rangle$ denotes cosine similarity. Accordingly, we ensure

samples that have the same classes and the same level of difficulty to stay close in the feature space. Moreover, while the anchor sample $(\mathbf{x}_i, y_i)$ and the $k$-most dissimilar samples from the opposite level of difficulty $(\mathbf{x}_i'', y_i'')$ are further apart because they are dissimilar in feature representations, our method pulls these samples together to encourage them to stay close. Consequently, we allow samples that have the same class to be tightly group together in the feature space. By our construction of positive sets, we enforce that the model is explicitly exposed to both easy and challenging samples which can help the model to adjust its confidence. Thus, we prevent the model from being biased toward some category, e.g., easy to learn, and in turn, we prevent the model from becoming too certain on its prediction. Hence, we increase the robustness of the model, leading to improved model calibration.

**Constructing Negative Sets**  We observe from Figure 1(c) that ambiguous samples are not completely separable along their classes. In addition, from Figure 1(b) we observe that while easy-to-learn samples are fairly clustered along class labels compared to ambiguous samples, there are still overlapping samples between class clusters (e.g., 'contradiction' and 'neutral'). In both cases, we aim to push them apart. Thus, for a sample (anchor) that belongs to class $c$, we construct its negative set with all samples that have different class than $c$ but have the same level of difficulty to the anchor. For example, if an anchor sample is ambiguous $(\mathbf{x}_i, y_i) \in \mathcal{D}_{ambig}$ and has class label $c$, we then construct the negative set $N(i)$ by selecting ambiguous samples $(\mathbf{x}_i', y_i') \in \mathcal{D}_{ambig}$ that have a different class label than the anchor (i.e., $y_i' \neq c$), and whose cosine similarities with $\mathbf{x}_i$ are higher than a pre-defined threshold $\tau$. By using a pre-defined threshold, we ensure that the selected ambiguous samples $(\mathbf{x}_i', y_i')$ and the ambiguous anchor $(\mathbf{x}_i, y_i)$ are pushed further apart. We set the threshold value $\tau$ relatively high ($\tau = 0.8$) to allow fairly similar samples to be selected. We formulate our negative set construction as follows:

$$N(i) = \{(\mathbf{x}_i', y_i') \in \mathcal{D}_{SameDiffic} : y_i' \neq c, \langle \mathbf{x}_i', \mathbf{x}_i \rangle > \tau\} \tag{2}$$

where $\mathcal{D}_{SameDiffic}$ denotes the same learning difficulty as the anchor, $\langle \cdot, \cdot \rangle$ denotes cosine similarity, and $\tau$ is a pre-defined threshold. Accordingly, we force the model to push samples that have different class labels further apart, and hence, to learn to better separate them. By our negative set construction, we specifically inform the model to push further apart samples that are similar in representations and learning difficulty but have different class labels. This allows the model to encounter hard negatives that are more challenging. Hence, we enhance the robustness of the model and its calibration.

**Supervised Contrastive Loss as a Regularizer**  After generating the positive and negative sets for each anchor (ambiguous or easy-to-learn) sample, the supervised contrastive loss becomes:

$$\mathcal{L}_{supCL} = \sum_{i=0}^{N} \frac{-1}{|P(i)|} \sum_{\mathbf{x}_p \in P(i)} \log \frac{e^{\langle \mathbf{x}_i, \mathbf{x}_p \rangle / T}}{\sum_{\mathbf{x}_b \in N(i)} e^{\langle \mathbf{x}_i, \mathbf{x}_b \rangle / T}} \tag{3}$$

where $T$ is a temperature scaling parameter, $\langle \cdot, \cdot \rangle$ refers to cosine similarity, $P(i)$ is the positive set, and $N(i)$ is the negative set for $\mathbf{x}_i$.

Our final training objective is calculated by a weighted sum of cross-entropy loss and supervised contrastive loss on the top 33% easy-to-learn and top 33% ambiguous samples as follows:

$$\mathcal{L} = \mathcal{L}_{ce} + \lambda \mathcal{L}_{supCL}$$

where $\lambda$ is a hyper-parameter. Our supervised contrastive loss term performs as a regularizer, hence, probabilities become smoother compared to only using cross-entropy loss which usually results in overly confident predictions. We evaluate its effect on model calibration in addition to accuracy for NLP tasks. We summarize our approach in Algorithm 1 in Appendix A.1.

## 4 EXPERIMENTS

### 4.1 TASKS AND DATASETS

**Sentence-pair tasks**  We evaluate our method on natural language inference (NLI), paraphrase identification, and commonsense reasoning. For NLI, we use Stanford Natural Language Inference

(SNLI) dataset Bowman et al. (2015) for in-domain evaluation. We use Multi-Genre NLI (MNLI) that captures NLI with diverse domains Williams et al. (2018) as SNLI's out-of-distribution (OOD) evaluation. For paraphrase identification, we use Quora Question Pairs (QQP) Iyer et al. (2017) as in-domain evaluation and TwitterPPDB (TPPDB) which determines whether sentence pairs from Twitter convey similar semantics when they share URLs Lan et al. (2017) as QQP's OOD evaluation. For commonsense reasoning, we test our method on Situations With Adversarial Generations (SWAG) aiming to choose the most plausible continuation of a sentence among four candidates Zellers et al. (2018), and use HellaSWAG, a dataset built using adversarial filtering Zellers et al. (2019) as SWAG's OOD evaluation.

**Single-sentence tasks**   We evaluate our method on movie review sentiment analysis using MR dataset Pang & Lee (2005) and opinion polarity classification using the MPQA dataset Wiebe et al. (2005). For MR's OOD evaluation, we use Customer Review (CR) test data Hu & Liu (2004). For MPQA's OOD evaluation, we use Pro-Con (PC) Ganapathibhotla & Liu (2008) test data.

## 4.2   EXPERIMENTAL SETUP

We use the BERT-base Devlin et al. (2019) classification model.   Training details and hyper-parameter settings can be found in Appendix A.3. We provide a detailed analysis of hyper-parameter selection for positive set construction (i.e., $k$) and negative set construction (i.e., $\tau$) in Appendix A.4. We evaluate the capability of our method to improve both predictive performance and model calibration. Hence, we use two metrics: (1) accuracy, and (2) expected calibration error (ECE) Guo et al. (2017). We provide a detailed definition of ECE in the Appendix. For each task, we train the model on the in-domain training set and evaluate its accuracy and ECE on in- and out-of-domain test sets.

## 4.3   BASELINE METHODS

**Supervised Learning**   We use the pre-trained BERT model Devlin et al. (2019) fine-tuned on each downstream task.

**Supervised Curriculum Learning (CurricLearn)**   is a learning strategy for training a model from easy samples to difficult ones. For this, we use the following difficulty measures: **(1)** the input lengths of training samples Nagatsuka et al. (2023) and **(2)** the level of difficulty obtained by using data maps (the top 33% easy and 33% ambiguous samples) Swayamdipta et al. (2020).

**Unsupervised Contrastive Learning**   To compare with unsupervised contrastive learning methods, we follow the downstream evaluation protocol by Robinson et al. (2021), with using BERT encoders and BERT classifiers. Our implementation details can be found in Appendix A.6. We use the following methods: **(1) MoCHi** Kalantidis et al. (2020) generates negative sets through MixUp Zhang et al. (2018) in the latent space.  The positive sets are constructed using adjacent sentences (before/after) of a given sentence in the BookCorpus; **(2) Contrastive Learning with Hard Negatives** Robinson et al. (2021) constructs negative sets by up-weighting the negative points that have a larger inner product and have different latent classes with the anchor. The positive pairs are constructed using adjacent sentences (before/after) of a given sentence in the BookCorpus; **(3) SimCSE** Gao et al. (2021) generates positive sets by independently sampling dropout masks of a given sample, and generates negative sets by selecting random samples in a given batch; and **(4) SimCSE++** Xu et al. (2023) extends SimCSE by generating negative sets without applying dropout masks in addition to combining dimension-wise contrastive learning objective.

**Supervised Contrastive Learning** We compare SupCL-GSS with the following existing supervised contrastive learning methods: **(1) SupCL** Gunel et al. (2020) combines supervised contrastive learning with the cross-entropy loss. Both positive and negative sets are constructed by only leveraging class label information in a given batch; and **(2) SupCL-Seq** Sedghamiz et al. (2021) produces positive sets not only by selecting samples that have the same class label in a given batch, but also by applying different dropout masks on samples that have the same class. Negative sets are constructed by selecting random samples that have different class labels in a given batch.

| | Acc | ECE | Acc | ECE | Acc | ECE | Acc | ECE | Acc | ECE |
|---|---|---|---|---|---|---|---|---|---|---|
| | | SNLI | | QQP | | SWAG | | MR | | MPQA |
| BERT | 90.04 | 2.54 | 90.27 | 2.71 | 79.40 | 2.49 | 86.34 | 4.69 | 88.25 | 8.56 |
| CurricLearn w/ length Nagatsuka et al. (2023) | 87.73 | 2.66 | 86.37 | 4.99 | 78.81 | 5.38 | 86.45 | 3.66 | 87.55 | 10.71 |
| CurricLearn w/ data maps | 89.07 | 6.29 | 88.45 | 6.78 | 78.45 | 5.68 | 86.55 | 5.14 | 87.15 | 8.46 |
| MoCHi Kalantidis et al. (2020) | 89.53 | 2.35 | 90.02 | 2.84 | 78.81 | 1.87 | 87.10 | 4.02 | 87.75 | 9.07 |
| CL w/ HN Robinson et al. (2021) | 90.06 | 6.07 | 90.54 | 6.64 | 79.37 | 2.14 | 86.75 | 4.47 | 88.17 | 8.23 |
| SimCSE Gao et al. (2021) | 90.07 | 3.54 | 89.93 | 5.27 | 79.58 | 1.96 | 86.60 | 3.45 | 87.60 | 9.62 |
| SimCSE++ Xu et al. (2023) | 89.65 | 2.48 | 89.64 | 4.32 | 79.16 | 2.03 | 86.88 | 2.46 | 87.70 | 9.85 |
| SupCL Gunel et al. (2020) | 89.78 | 2.50 | 90.32 | 2.39 | 78.82 | 6.81 | 86.95 | 3.06 | 87.34 | 6.39 |
| SupCL-Seq Sedghamiz et al. (2021) | 90.28 | 3.51 | 89.92 | **2.27** | 78.59 | 4.22 | 87.14 | 4.88 | 88.06 | 8.08 |
| **SupCL-GSS (Ours)** | **90.44**[†] | **1.31**[†] | **90.88** | 2.46[†] | **79.69**[†] | **1.54**[†] | **87.25**[†] | **1.74**[†] | **88.93**[†] | **6.11**[†] |
| | | MNLI | | TwitterPPDB | | HellaSWAG | | CR | | PC |
| BERT | 73.52 | 7.09 | 87.63 | 8.51 | 34.48 | 12.62 | 85.49 | 3.46 | 85.02 | 10.74 |
| CurricLearn w/ length Nagatsuka et al. (2023) | 73.27 | 3.57 | 86.94 | 10.76 | 34.17 | 18.64 | 86.53 | 1.59 | 81.45 | 16.18 |
| CurricLearn w/ data maps | 71.75 | 16.09 | 87.43 | 10.22 | 34.65 | 13.35 | 82.65 | 4.71 | 83.66 | 10.55 |
| MoCHi Kalantidis et al. (2020) | 73.09 | 5.12 | 87.45 | 8.63 | 34.71 | 11.02 | 82.35 | 5.35 | 79.13 | 5.19 |
| CL w/ HN Robinson et al. (2021) | 73.32 | 10.76 | 86.59 | 9.48 | 34.49 | 10.01 | 85.39 | 2.48 | 82.03 | 5.45 |
| SimCSE Gao et al. (2021) | 72.41 | 4.54 | 86.71 | 9.59 | 33.88 | 14.54 | 85.46 | 1.88 | 84.82 | **4.76** |
| SimCSE++ Xu et al. (2023) | 73.26 | 3.52 | 86.92 | 8.88 | 34.16 | 13.38 | 85.59 | 1.93 | 84.14 | 8.72 |
| SupCL Gunel et al. (2020) | 73.55 | 9.09 | 87.19 | 9.43 | 34.27 | 19.20 | 86.63 | 3.55 | 85.63 | 6.46 |
| SupCL-Seq Sedghamiz et al. (2021) | 72.11 | 12.54 | **87.71** | 7.46 | 34.51 | 10.05 | 85.87 | 4.46 | 86.66 | 8.49 |
| **SupCL-GSS (Ours)** | **74.01**[†] | **2.76**[†] | 87.52 | 6.24 | **35.12**[†] | **9.15**[†] | **87.25**[†] | 1.54 | 86.93 | 6.05 |

Table 1: Accuracy (Acc) and Expected Calibration Error (ECE) in percentage on in-domain (top) and out-of-domain (bottom) comparing SupCL-GSS with baseline methods. Lower ECE implies better-calibrated models. Bold text shows the best Acc and ECE. †: SupCL-GSS improves the best baseline at p<0.05 with paired t-test.

| | Acc | ECE | Acc | ECE | Acc | ECE | Acc | ECE | Acc | ECE |
|---|---|---|---|---|---|---|---|---|---|---|
| | | SNLI | | QQP | | SWAG | | MR | | MPQA |
| SupCL-GSS (Ours) | **90.44** | **1.31** | **90.88** | **2.46** | **79.69** | **1.54** | **87.25** | **1.74** | **88.93** | 6.11 |
| SupCL-GSS on 25% Easy & 25% Ambig | 89.04 | 2.84 | 89.56 | 3.11 | 78.56 | 3.55 | 86.07 | 2.44 | 87.60 | 7.84 |
| SupCL-GSS on 50% Easy & 50% Ambig | 89.76 | 2.31 | 90.08 | 3.42 | 79.51 | 3.28 | 86.75 | 2.87 | 87.23 | 8.69 |
| SupCL-GSS on 33% Bottom Confidence & 33% Easy | 90.24 | 2.05 | 89.86 | 2.87 | 79.10 | 4.45 | 86.23 | 2.31 | 87.73 | **5.22** |
| SupCL-GSS on 33% Bottom Confidence & 33% Ambig | 87.70 | 5.66 | 86.05 | 3.56 | 78.33 | 5.02 | 81.05 | 6.17 | 86.07 | 8.68 |
| | | MNLI | | TwitterPPDB | | HellaSWAG | | CR | | PC |
| SupCL-GSS (Ours) | **74.01** | **2.76** | **87.52** | **6.24** | **35.12** | **9.15** | **87.25** | **1.54** | **86.93** | **6.05** |
| SupCL-GSS on 25% Easy & 25% Ambig | 73.43 | 5.81 | 86.95 | 8.03 | 34.59 | 12.22 | 83.36 | 9.43 | 82.38 | 6.44 |
| SupCL-GSS on 50% Easy & 50% Ambig | 73.85 | 4.14 | 87.06 | 7.16 | 34.75 | 16.82 | 84.51 | 6.84 | 83.86 | 13.24 |
| SupCL-GSS on 33% Bottom Confidence & 33% Easy | 73.19 | 8.42 | 87.11 | 8.71 | 34.03 | 12.27 | 84.12 | 6.83 | 87.72 | 9.39 |
| SupCL-GSS on 33% Bottom Confidence & 33% Ambig | 70.28 | 11.19 | 85.93 | 8.63 | 33.26 | 10.75 | 73.63 | 8.06 | 81.77 | 11.49 |

Table 2: The results of SupCL-GSS using samples identified by different levels of difficulty on in-domain (top) and out-of-domain (bottom) settings.

## 5 RESULTS AND ANALYSIS

### 5.1 MAIN RESULTS

In Table 1, we show the comparison of SupCL-GSS with baseline methods. We report the evaluation method of unsupervised contrastive learning methods in Appendix A.6. Remarkably, SupCL-GSS yields performance improvement and lower calibration errors compared to all baseline methods on both sentence-pair tasks and single-sentence tasks, and on both in- and out-of-domain data in general. SupCL-GSS surpasses all supervised curriculum learning baselines, highlighting the advantage of its using different levels of difficulty samples in hard positive and hard negative constructions over using them in the pre-defined order (i.e., from easy to difficult ones). Furthermore, SupCL-GSS achieves better accuracy and lower ECEs compared with the existing unsupervised and supervised contrastive learning because SupCL-GSS leverages both hard positives and hard negatives.

### 5.2 ANALYSIS

Here, we first explore different ratios of easy-to-learn and ambiguous samples (i.e., 25% and 50% rather than 33%). We also analyze the impact of other samples from the training data on SupCL-GSS. Second, we perform an ablation where we remove each component of SupCL-GSS one at a time. Last, we study the impact of batch size Chen et al. (2020) to identify hard/challenging positive and negative sets from larger and diverse batches.

| | Acc | ECE | Acc | ECE | Acc | ECE | Acc | ECE | Acc | ECE |
|---|---|---|---|---|---|---|---|---|---|---|
| | SNLI | | QQP | | SWAG | | MR | | MPQA | |
| SupCL-GSS | **90.44** | **1.31** | **90.88** | 2.46 | **79.69** | **1.54** | **87.25** | **1.54** | **88.93** | 6.11 |
| w/o PosSet | 89.34 | 1.46 | 90.43 | 3.29 | 78.87 | 1.74 | 86.55 | 2.54 | 87.81 | 6.84 |
| w/o NegSet | 90.10 | 2.01 | 90.72 | 2.51 | 78.34 | 1.96 | 85.49 | 2.62 | 88.36 | 6.27 |
| w/o Learning Difficulty | 89.94 | 1.54 | 90.05 | **1.96** | 79.14 | 2.58 | 84.60 | 3.47 | 87.78 | **4.72** |
| w/o CosSim | 89.91 | 1.38 | 90.14 | 2.22 | 78.17 | 2.06 | 86.50 | 3.21 | 87.75 | 6.02 |
| | MNLI | | TwitterPPDB | | HellaSWAG | | CR | | PC | |
| SupCL-GSS | **74.01** | **2.76** | **87.52** | **6.24** | **35.12** | **9.15** | **87.25** | **1.54** | **86.93** | 6.05 |
| w/o PosSet | 73.55 | 2.81 | 87.28 | 7.80 | 35.07 | 10.05 | 85.94 | 2.91 | 85.01 | 6.38 |
| w/o NegSet | 72.69 | 6.18 | 87.45 | 7.33 | 34.79 | 12.47 | 86.48 | 10.31 | 86.14 | **5.51** |
| w/o Learning Difficulty | 73.62 | 3.53 | 86.69 | 8.39 | 34.89 | 13.46 | 81.95 | 2.38 | 86.24 | 5.69 |
| w/o CosSim | 73.39 | 2.96 | 87.09 | 8.37 | 34.73 | 11.71 | 85.40 | 1.55 | 83.18 | 6.08 |

Table 3: The results of SupCL-GSS removing different parts on in-domain (top) and out-of-domain (bottom) settings.

| | Acc | ECE | Acc | ECE | Acc | ECE |
|---|---|---|---|---|---|---|
| | SNLI | | QQP | | SWAG | |
| SupCL-GSS | **90.44** | **1.31** | **90.88** | **2.46** | **79.69** | **1.54** |
| BS = 128 | 90.34 | 2.06 | 90.62 | 3.33 | 79.46 | 1.87 |
| | MNLI | | TwitterPPDB | | HellaSWAG | |
| SupCL-GSS | **74.01** | **2.76** | **87.52** | **6.24** | **35.12** | **9.15** |
| BS = 128 | 73.36 | 2.81 | 87.09 | 7.35 | 34.59 | 10.02 |

Table 4: The results of SupCL-GSS with larger batch size on SNLI, QQP, and SWAG on in-domain (top) and out-of-domain (bottom) settings.

**Using Different Ratio of Easy and Ambiguous Samples**   To leverage the most representative samples in training data, we choose the top 33% easy-to-learn and the top 33% ambiguous after ranking samples according to training dynamics. Here, we explore different ratios, which are 25% and 50%, and show results in Table 2 (*25% Easy & 25% Ambig* and *50% Easy & 50% Ambig*). We observe that using a 25% ratio degrades both accuracy and ECE, which implies that using only 25% easy and 25% ambiguous samples limits access to all the representative samples (i.e., does not ensure enough diversity and challenging nature in the samples). Furthermore, we observe that selecting more easy-to-learn and ambiguous samples than 33% degrades accuracy and worsens ECE, in both in- and out-of-domain settings. We posit that this is because lowering the threshold too much on easy-to-learn samples blurs the line between easy and ambiguous samples while lowering the threshold too much on ambiguous samples introduces potentially erroneous samples from the data which degrades performance. To this end, we conclude that selecting the top 33% easy-to-learn and the top 33% ambiguous samples is a reasonable choice. We also analyze the impact of selecting samples from the bottom 33% ranked by confidence and pairing them with either 33% easy or 33% ambiguous samples (*33% Bottom Confidence & 33% Easy* and *33% Bottom Confidence & 33% Ambig*). We observe performance degradation and increase in calibration errors in both cases. We hypothesize that this is because of potential erroneous labels that exist in the 33% Bottom Confidence samples as discussed also in Swayamdipta et al. (2020). Accordingly, we conclude that using 33% easy and 33% ambiguous samples is a reasonable design choice in SupCL-GSS.

**Ablation Analysis for SupCL-GSS**   In Table 3, we report the results of SupCL-GSS after removing each component one at a time: (a) removing positive set construction from Eq. (1) (*w/o PosSet*). For this, we construct the positive set by using class label only; (b) removing negative set construction from Eq. (2) (*w/o NegSet*). We construct the negative set by using class label only; (c) without using the learning difficulty where easy-to-learn and ambiguous samples are combined and randomly split into two sets for positive and negative sets constructions (*w/o Learning Difficulty*); and (d) without using the cosine similarity in both positive and negative sets constructions (*w/o CosSim*). We observe performance degradation and an increase in ECEs after removing each component, suggesting that all components in SupCL-GSS contribute to the final performance and calibration.

**Batch Size Analysis**   To explore the effect of increasing the batch size on SupCL-GSS, we quadruple the batch size (original batch size of 32) for SNLI/SWAG, and double the batch size (original

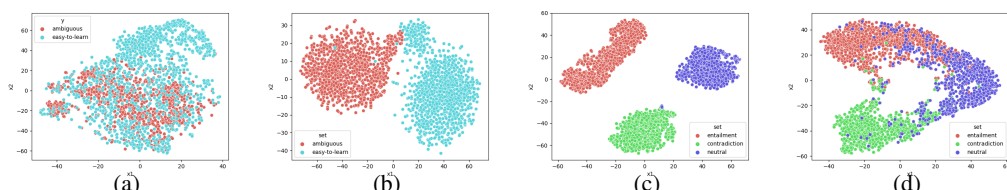

Figure 2: t-SNE visualization of 1,000 samples of (a) easy-to-learn and ambiguous SNLI *entailment* label train data on SupCL-GSS, and (b) SupCL-GSS without using the level of difficulty. We also visualize (c) the top 33% easy-to-learn and (d) ambiguous all class labels of SNLI train data.

batch size of 64) for QQP. We show these results in Table 4. Interestingly, we observe similar (or worse) performance. We conclude that SupCL-GSS enables informative selection when constructing positive and negative sets even when using a smaller batch size, and enlarging the batch size has a limited impact.

## 6 T-SNE VISUALIZATION

When designing the construction of the positive and negative sets, we hypothesize the following:

1. Our generated positive sets allow samples that have different level of difficulty but belong to the same class to stay close together in the feature space.

2. Our generated negative sets allow samples that have the same level of difficulty but belong to different classes to stay further apart in the feature space.

Specifically, according to hypothesis (1), we encourage easy-to-learn and ambiguous samples that share the same class to be uniformly distributed in the feature space regardless of their difficulty level. According to hypothesis (2), we ensure ambiguous or easy-to-learn samples that have different class labels to be separable along their classes.

To validate our hypotheses, we plot the hidden representations of samples in the feature space using t-SNE, which are obtained by SupCL-GSS. Specifically, in Figure 2 we plot the same 1,000 easy-to-learn and ambiguous SNLI train data (as in Figure 1). Figure 2a shows the 1,000 samples from the top 33% easy-to-learn and top 33% ambiguous SNLI 'entailment' class train data. We observe that samples are no longer separable into easy-to-learn and ambiguous, but form a tighter cluster (compared with Figure 1a). In Figure 2b, we plot the samples used in Figure 2a, but without specifying (or leveraging) the level of difficulty where we combine easy-to-learn and ambiguous samples and randomly split them into two sets to apply our method of constructing positive and negative sets. Consequently, we allow positive and negative sets to have pairs of: (1) easy-to-learn and easy-to-learn, (2) easy-to-learn and ambiguous, and (3) ambiguous and ambiguous samples. We observe SupCL-GSS without specifying the level of difficulty separates easy-to-learn and ambiguous samples even more clearly. This implies our way of specifying the level of difficulty of samples is a required component in SupCL-GSS. These results support hypothesis (1). Moreover, in Figures 2c and 2d, we plot 1,000 samples from the top 33% easy-to-learn (2c) and ambiguous (2d) of SNLI train data belonging to all classes (i.e., 'contradiction', 'entailment', and 'neutral'), respectively. We observe that samples are better separable along their classes in each category (as desirable) compared to Figures 1b and 1c, which supports hypothesis (2).

## 7 CONCLUSION

We proposed SupCL-GSS, supervised contrastive learning with enhancing selection for constructing hard positive and negative sets guided by both samples' learning difficulty and class labels, to improve the generalization of learned representations within pre-trained language models for text classification, achieving better accuracy and mitigating error in calibration. We empirically validate that SupCL-GSS achieves statistically significant improvements over the best baseline on various text classification tasks, with better accuracy and lower calibration errors on various NLP tasks on in- and out-of-domain settings compared to a wide range of competitive baselines.

# A APPENDIX

## A.1 ALGORITHM

We summarize our approach in Algorithm 1.

---

**Algorithm 1** : Proposed Approach

---

**Require:** Top 33% Easy-to-learn set,
$\qquad \mathcal{D}_{easy} = \{(x_i, y_i)\}_{i=1,\cdots,n}$
$\qquad$ Top 33% Ambiguous set,
$\qquad \mathcal{D}_{ambig} = \{(x_i, y_i)\}_{i=1,\cdots n}$
$\qquad$ Pre-trained Language Model $f$
0: **for** $k := 0$ to T **do**
0: $\quad Total\_Loss = 0$
0: $\quad$ **for** $\forall i, (x_i, y_i) \in \mathcal{D}_{easy} \cup \mathcal{D}_{ambig}$ **do**
0: $\qquad P(i), N(i) \leftarrow \emptyset$
0: $\qquad$ **while** $|P(i)|! = BatchSize - k$ **do**
0: $\qquad\quad$ Select samples from $\mathcal{D}_{SameDifficulty}$ that have the same label as $x_i$ and add them to $P(i)$
0: $\qquad$ **end while**
0: $\qquad$ Find the $k$ most dissimilar instances $(x_i'', y_i'')$ from $\mathcal{D}_{OppositeDifficulty}$ and add them to $P(i)$
0: $\qquad$ **while** $|N(i)|! = BatchSize$ **do**
0: $\qquad\quad$ Randomly select sample $(x_j, y_j)$ from $\mathcal{D}_{SameDifficulty}$
0: $\qquad\quad$ **if** $CosSim(f(x_j), f(x_i)) > \tau$ and $y_j \neq y_i$
0: $\qquad\quad$ **then** $N(i) \leftarrow (x_j, y_j) \cup N(i)$
0: $\qquad$ **end while**
0: $\qquad$ Calculate Supervised Contrastive Loss $L_{supCL}$ using Eq. (3)
0: $\qquad$ Calculate Cross Entropy Loss $\mathcal{L}_{ce}$
0: $\qquad Loss = \mathcal{L}_{ce} + \lambda \mathcal{L}_{supCL}$
0: $\quad$ **end for**
0: $\quad Total\_Loss \leftarrow Total\_Loss + Loss$
0: $\quad$ Update the model weights
0: **end for**=0

---

## A.2 SUPCL-GSS ON ROBERTA-LARGE

To facilitate future research and replication of results, we used the relatively lightweight BERT-base-uncased as our backbone model. However, our method is flexible and can be used with any pre-trained language model such as RoBERTa-large. Hence, we perform experiments of SupCL-GSS using RoBERTa-large and compare these results with vanilla RoBERTa-large on SNLI (in-domain) and MNLI (out-of-domain) in Table 5. we observe our method achieves not only higher accuracy but also lower calibration error (ECE) compared to the baseline vanilla BERT, which proves the effectiveness of the proposed method.

| | Acc | ECE | Acc | ECE |
| --- | --- | --- | --- | --- |
| | SNLI | | MNLI | |
| RoBERTa-large | 91.04 | 2.01 | 78.86 | 4.62 |
| SupCL-GSS on RoBERTa-large | **92.18**[†] | **1.88**[†] | **79.83**[†] | **3.07**[†] |

Table 5: The results of SupCL-GSS on RoBERTa-large on SNLI (in-domain) and MNLI (out-of-domain). †: our method improves the the best baseline at $p < 0.05$ with paired t-test.

## A.3 TRAINING DETAILS

In our experiments, we use the *bert-base-uncased* model with a task-specific fully-connected classification layer on top All hyper-parameters are estimated on the validation set of each task. Specifically, we estimate hyper-parameters via a grid search over combinations. We use the following range of values to determine the best hyper-parameters: batch size 1,4,8,16,32,64,128, learning rate (1e-3, 2e-3, 1e-4, 2e-4, 1e-5, 2e-5), temperature scaling $T$ on supervised contrastive

| Dataset | Train | Test |
|---|---|---|
| SNLI | 549,368 | 4,923 |
| MNLI | 392,702 | 4,907 |
| QQP | 363,871 | 20,217 |
| TwitterPPDB | 46,667 | 5,060 |
| SWAG | 73,547 | 10,004 |
| HellaSWAG | 39,905 | 5,021 |
| MR | 8,662 | 2,000 |
| CR | 1,775 | 2,000 |
| MPQA | 8,606 | 2,000 |
| Pro-Con (PC) | 41,877 | 4,000 |

Table 6: The statistics of in-domain and out-of-domain datasets.

| | SNLI | | QQP | | SWAG | | MR | | MPQA | |
|---|---|---|---|---|---|---|---|---|---|---|
| | Acc | ECE | Acc | ECE | Acc | ECE | Acc | ECE | Acc | ECE |
| $k = 1$ (SupCL-GSS, Ours) | **90.63** | **1.65** | **89.97** | 3.14 | **79.59** | 2.03 | **87.64** | 3.23 | **98.37** | **0.83** |
| $k = 2$ | 90.42 | 1.94 | 88.71 | 5.41 | 78.77 | **1.77** | 87.46 | 3.31 | 98.05 | 1.61 |
| $k = 5$ | 90.14 | 1.83 | 88.65 | 3.88 | 79.06 | 3.48 | 87.43 | 4.19 | 98.02 | 1.39 |
| $k = 16$ | 90.43 | 2.01 | 89.64 | 3.68 | 78.29 | 2.15 | 87.15 | 3.91 | **98.37** | 1.21 |

Table 7: Acc and ECE in percentage on in-domain validation data for selecting the hyper-parameter $k$ in positive set construction for SupCL-GSS.

loss [1e-2, 5e-1]. The model is fine-tuned with a maximum of 3 epochs, batch size of 32 for SNLI/SWAG/MR/MPQA and 64 for QQP, a learning rate of 2e-5, gradient clip of 1.0, and no weight decay. We set temperature scaling $T$ on supervised contrastive loss as 0.05/0.1/0.01/0.01/0.01 for SNLI/QQP/SWAG/MR/MPQA. We set a weight $\lambda = 0.1$ on the final training loss. In generating positive sets, we select the most dissimilar sample from the other level of difficulty set (i.e., $k = 1$) for all tasks. Finally, all experiments are conducted on a single NVIDIA RTX A6000 48G GPU with the total time for fine-tuning all models being under 24 hours. Note that our implementation utilizes easy-to-learn and ambiguous data loaders separately and does selection using mini-batch for computational efficiency. For sentence-pair task datasets, we follow the published train/validation/test split by Desai & Durrett (2020). For single-sentence task datasets, we used the publicly released train/test split where we randomly selected 2,000 samples as a testing set on MR/MPQA and left them out from training. We show the statistics of the datasets in Table 6.

## A.4 HYPER-PARAMETER SELECTION ON $\tau$ AND $k$

**Hyper-parameter $k$** We estimate the positive set construction hyper-parameter $k$ via a search in $k = \{1, 2, 5, 16\}$ on the in-domain validation data for each task and selected the value of $k$ that maximizes accuracy. We then used the best hyper-parameter value on the test set. We show the accuracy and Expected Calibration Error (ECE) on all in-domain validation datasets for $k = 1, 2, 5, 16$ in Table 7. We observe from these results that using $k = 1$ achieves the best performance in general whereas other values of $k$ either perform the same or yield a slight drop in performance. Thus, based on these results, we set $k = 1$ to report the test data results.

**Hyper-parameter $\tau$** We estimate the negative set construction hyperparameter $\tau$ via a search in $\tau = \{0.5, 0.7, 0.8, 0.9\}$ on the in-domain validation data for each task and select the value of $\tau$ that maximizes accuracy. We aim to set $\tau$ to a relatively high value (i.e., no smaller than 0.5) to ensure we select fairly similar feature representations to generate hard negatives. We then use the best hyper-parameter $\tau$ value on the test set. We show the accuracy and Expected Calibration Error (ECE) on all in-domain validation datasets for $\tau = \{0.5, 0.7, 0.8, 0.9\}$ in Table 8. We observe from these results that using $\tau = 0.8$ achieves the best performance in general whereas other values of $\tau$ either perform similarly or yield a slight drop in performance. Thus, based on these results, we set $\tau = 0.8$ to report the test data results.

| | SNLI | | QQP | | SWAG | | MR | | MPQA | |
|---|---|---|---|---|---|---|---|---|---|---|
| | Acc | ECE | Acc | ECE | Acc | ECE | Acc | ECE | Acc | ECE |
| $\tau = 0.5$ | 90.39 | 1.91 | 88.72 | 3.23 | 79.07 | 3.67 | 87.25 | 3.49 | 97.56 | 1.63 |
| $\tau = 0.7$ | 90.04 | 2.17 | 89.66 | 3.65 | 79.12 | 4.85 | 87.06 | 3.68 | 98.13 | 0.98 |
| $\tau = 0.8$ (SupCL-GSS, ours) | **90.63** | **1.65** | **89.97** | 3.14 | **79.59** | **2.03** | **87.64** | 3.23 | 98.37 | **0.83** |
| $\tau = 0.9$ | 90.26 | 1.78 | 88.61 | **3.11** | 79.21 | 4.03 | 87.29 | 2.27 | 98.14 | 1.01 |

Table 8: Acc and ECE in percentage on in-domain validation data for selecting the hyper-parameter $\tau$ in negative set construction for SupCL-GSS.

## A.5 CALIBRATION METRIC: EXPECTED CALIBRATION ERROR (ECE)

A model is perfectly calibrated when its confidence estimate $\hat{p}$ matches the true probability (accuracy), such that $\mathbb{P}(\hat{y} = y|\hat{p}) = \hat{p}$ Naeini et al. (2015); Guo et al. (2017); Desai & Durrett (2020). To approximate this empirically, the probability range is divided into a set number of bins where each bin $b_m$ contains predicted probabilities within a specific interval. The expected calibration error (ECE) is then computed by taking a weighted average of the differences between each bin's accuracy and confidence, as follows:

$$\text{acc}(b_m) = \frac{1}{|b_m|} \sum_{i \in b_m} \mathbb{1}(\hat{y}_i = y_i)$$

$$\text{conf}(b_m) = \frac{1}{|b_m|} \sum_{i \in b_m} \hat{p}_i$$

$$\text{ECE} = \sum_{m=1}^{M} \frac{|b_m|}{N} |\text{acc}(b_m) - \text{conf}(b_m)|$$

where $N$ is the total number of predictions.

## A.6 UNSUPERIVSED CONTRASTIVE LEARNING BASELINE IMPLEMENTATION

The main purpose of unsupervised contrastive learning baseline methods is to pre-train representations (i.e., features) that can be transferred to downstream tasks by fine-tuning. We test unsupervised contrastive learning baseline methods on learning representations of sentences using the quick thoughts (QT) vectors framework introduced by Logeswaran & Lee (2018). Specifically, for each baseline, we first train BERT encoders following their proposed methods using BookCorpus Kiros et al. (2015). Afterward, we obtain feature representations of sentences from the trained BERT encoders. We then train BERT classifiers on top of the generated feature representations for each task. We use the same hyperparameter when further fine-tuning BERT classifier as our proposed method.

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
