# OpenReview forum: "SupCL-GSS: Supervised Contrastive Learning with Guided Sample Selection"
_ICLR.cc/2026/Conference — Submitted to ICLR 2026_

### Official Review · Reviewer_9TkM · 2025-10-22

**Soundness:** 3
**Presentation:** 3
**Contribution:** 2
**Rating:** 4
**Confidence:** 3

**Summary:**

This paper proposes SupCL-GSS, a supervised contrastive learning method with a new training sample selection strategy, to improve the downstream model's performance when trained on selected positive and negative samples. The training sample selection strategy selects positive training samples by pairing an anchor example with sample-class, same-difficulty, and most dissimilar same-class examples, and negative samples by pairing same-difficulty and different-class instances. This process finds hard positives, which the authors argue are equally important as hard negatives. The difficulty is estimated by data maps, a method from Swayamdipta et al. (2020) that uses model-predicted confidence and consistency. Empirical results show that the proposed method can find good pairs of positive and negative samples, leading to effective contrastive learning and surpassing several baselines.

**Strengths:**

The data selection strategy uses data maps, which approximate the training dynamics by observing the model's confidence over epochs, to model learning difficulty and construct hard positive and negative pairs for more effective contrastive learning. This source of learning difficulty in finding data pairs is interesting and may inspire future works. Based on this idea, the paper presents a coherent learning scheme that future work could build on. The paper presents evaluations of both accuracy and calibration error (ECE), providing further insight into the idea's intuition and effectiveness.

**Weaknesses:**

My main concern is with the amount of contribution to the community. The paper proposes the selection of hard positives and hard negatives for contrastive learning, which is not new and has been discussed in previous works, albeit not necessarily directly in a pure NLP-based contrastive learning setting. Hard negatives are very well known in the ML community, and in NLP, have been widely discussed in RAG training, etc. Hard positives have also been similarly discussed, for example, in [1]. The real difference lies in how to select the difficulty, but the difficulty metric SupCL-GSS used is based on, and almost identical to, the one in the original 2020 paper on data maps.

[1] FaceNet: A Unified Embedding for Face Recognition and Clustering, Schroff et al. 2015.

**Questions:**

1. Is the proposed method trained on the same data as the baselines in the report experiments? If not, I think it would make sense to compare baselines in the same supervision settings.
2. Why are the MPQA accuracies in Table 8 much higher than those in Table 1?
3. What is the number of runs with different random seeds for the statistical tests?

---

### Official Review · Reviewer_ycnJ · 2025-10-30

**Soundness:** 1
**Presentation:** 2
**Contribution:** 2
**Rating:** 2
**Confidence:** 3

**Summary:**

This paper proposes Supervised Contrastive Learning with Guided Sample Selection (SupCL-GSS), a fine-tuning framework for pre-trained language models that incorporates sample difficulty into supervised contrastive learning. Motivated by the observation that “easy-to-learn” and “ambiguous” samples occupy distinct regions in representation space, the authors leverage this notion of learning difficulty to construct hard positives and hard negatives. SupCL-GSS is applied as an additional regularization term during fine-tuning, combining cross-entropy and supervised contrastive loss. Experiments across multiple NLP classification benchmarks demonstrate consistent improvements in both accuracy and model calibration over existing fine-tuning and contrastive learning baselines.

**Strengths:**

* The motivation is grounded in the empirical observation presented in Figure 1, where the authors visualize training samples based on their learning dynamics—confidence and variability computed following previous work. The figure shows that easy-to-learn and ambiguous samples occupy distinct regions in the representation space: easy samples form compact, class-specific clusters, whereas ambiguous samples are more dispersed and tend to overlap across class boundaries.

* Building on this insight, the authors propose Supervised Contrastive Learning with Guided Sample Selection (SupCL-GSS), which incorporates learning difficulty into the construction of contrastive pairs. The method introduces hard positives (same label but different difficulty) and hard negatives (different label but similar difficulty) to encourage the desired representation structure, as illustrated in Figure 2.

* The authors test their method on multiple NLP benchmarks spanning sentence-pair and single-sentence classification tasks, which provides a reasonable coverage of standard datasets.

**Weaknesses:**

### 1. Limitations of Experimental Evidence
The experiments do not convincingly demonstrate that incorporating learning difficulty into contrastive learning is either sufficient or necessary for improving representation quality or model calibration. The reported gains are small, unstable across hyperparameters, and lack statistical validation, making it unclear whether the proposed mechanism—rather than confounding factors—actually drives the improvements.

According to the authors, the ideal representation should satisfy two properties:
(1) positive sets encourage samples of different difficulty but the same class to cluster closely, and
(2) negative sets encourage samples of the same difficulty but different classes to remain apart.

**Sufficiency.**
* In the main results (Table 1), the reported accuracies are largely comparable across methods, and several contrastive learning baselines even underperform the simplest BERT fine-tuning baseline. This raises concerns about whether the baselines are implemented and tuned fairly, and whether the reported improvements are statistically meaningful.
* The paper claims significance (“SupCL-GSS improves the best baseline at *p* < 0.05 with paired *t*-test”), but it is unclear how this test was conducted. Since no standard deviations or multiple-run averages are reported, the reliability of this claim is questionable. Reporting mean ± standard deviation across multiple random seeds and showing training dynamics with confidence intervals would provide stronger evidence of stability and robustness.
* The hyperparameter sensitivity analysis (for $k$ and $\tau$) appears somewhat unstable. For example, the results for $k$ show that the method performs best when $k=1$, with only small and inconsistent changes as $k$ increases.

**Necessity.**
Although the authors hypothesise that aligning cross-difficulty positives and separating same-difficulty negatives improves representations, the paper does not empirically test whether these properties are *necessary* for the observed gains. For instance, the authors could have performed intervention-style analyses—systematically varying or even inverting these design choices—and observed how performance and calibration change while keeping other factors fixed. Showing a curve that reflects the effect of different levels of such interventions would also help visualise how strongly the proposed mechanism influences the outcome. Without such controlled experiments, it remains unclear whether the proposed mechanism is truly responsible for the improvements. This kind of control study is crucial for the scientific justification.


### 2. Lack of theoretical justification
There are several theoretical perspectives on what constitutes a good representation in contrastive learning, such as the analysis in *Tian et al., “What Makes for Good Views for Contrastive Learning?” (NeurIPS 2020)*. Establishing a clearer connection between the authors’ notion of an “ideal representation” and this line of theoretical work would make the paper’s claims more convincing and help situate the proposed mechanism within existing understanding of contrastive objectives.

**Questions:**

* The paper seems to inconsistently use `\citep` and `\citet` in several places (e.g., line 32). Please check and correct the citation formatting throughout.
* In the Table 1 caption, the notation “p¡0.05” appears to be a typographical error — it should be “p < 0.05.”
* Line 314: “SimCSE and the \citet” are concatenated without spacing, which likely indicates a missing space or incorrect citation command.

---

### Official Review · Reviewer_i6qo · 2025-10-31

**Soundness:** 3
**Presentation:** 3
**Contribution:** 2
**Rating:** 4
**Confidence:** 4

**Summary:**

This paper proposes SupCL-GSS, a supervised contrastive learning approach that leverages data maps to guide sample selection. The method constructs hard positives and hard negatives based on learning difficulty and class labels, aiming to improve both performance and calibration. Results show consistent gains on several NLP benchmarks.

**Strengths:**

* The idea of combining learning dynamics with contrastive learning is novel and intuitively appealing.

* Strong experimental results with extensive comparisons and ablations.

* The calibration analysis (ECE) is a nice addition that many contrastive papers ignore.

* The writing is good and easy to follow.

**Weaknesses:**

* The method depends on an additional fine-tuning stage to obtain sample difficulty, which might be costly.

* Choosing 33% as the easy/ambiguous threshold feels heuristic.Table 2 shows that using 25% as threshold causes a significant performance degradation, which make me worry about the hyperparameter sensitive of the proposed method.

* The choosed baselines seems to be too weak. For example, the baselines in table1 can hardly beat a naive BERT. Why?

**Questions:**

Is there any overlap between top33% easy samples and top33% ambiguous samples? Since they are selected based on different metrics (confidence vs variability), I'm wondering if there is any overlap?

---

### Official Review · Reviewer_LW6L · 2025-11-01

**Soundness:** 3
**Presentation:** 3
**Contribution:** 2
**Rating:** 4
**Confidence:** 3

**Summary:**

This paper proposes SupCL-GSS (Supervised Contrastive Learning with Guided Sample Selection), a method that uses training dynamics / data maps (confidence and variability per sample) to split training data into easy-to-learn and ambiguous subsets, and then constructs hard positives and hard negatives for supervised contrastive learning conditioned on (1) class label and (2) learning difficulty. Positives contain (a) same-class, same-difficulty samples and (b) the k most dissimilar same-class samples from the opposite difficulty set; negatives are same-difficulty, different-class samples whose cosine similarity to the anchor exceeds a threshold τ. SupCL-GSS is used as a regularizer (cross-entropy + λ·supCL) and evaluated on several sentence-pair and single-sentence classification tasks (SNLI/MNLI, QQP/TwitterPPDB, SWAG/HellaSWAG, MR/CR, MPQA/PC). The method reports consistent accuracy and expected calibration error (ECE) improvements over multiple supervised and contrastive baselines, ablation studies, and some visualizations (t-SNE).

**Strengths:**

1. Clear motivation: data-map notion maps well to constructing meaningful hard sets; t-SNE visualizations support intuition.

2. Comprehensive experimental sweep across in-domain and out-of-domain datasets, and both sentence-pair and single-sentence tasks.

3. Ablation studies and hyperparameter searches (k, τ, ratio) are thorough.

4. Shows gains in accuracy and calibration (ECE), the latter being less commonly targeted.

5. Simple to implement on top of existing finetuning pipelines and flexible to different PLMs (BERT, RoBERTa result shown).

**Weaknesses:**

1.Two-stage procedure (precompute data maps by fine-tuning, then re-train with supCL-GSS) increases training complexity and may introduce bias; runtime/cost analysis is missing.

2.Sensitivity to the choice of “top 33%” — authors tried 25%/50% but rationale for 33% still feels heuristic; more principled selection (or adaptive threshold) would be preferable.

3.Potential dataset / label-noise sensitivity — limited analysis on datasets with noisy annotations or heavy class imbalance.

4.Sparse reporting of variance: many tables show single numbers with † for significance but standard deviations over multiple runs are missing for several entries.

5.Generality: method is tailored to classification; claim about general representation improvements would need transfer evaluations (e.g., probing or few-shot tasks).

**Questions:**

1.How many epochs / what exact procedure did you use to compute the per-sample confidence/variability? Does using a different random seed or early-stopping change the partitioning significantly?

2.What is the wall-clock and GPU cost of the two-stage pipeline relative to baseline finetuning? Can the data maps be computed using fewer epochs or cheaper proxies?

3.How robust is SupCL-GSS to label noise? If a fraction of labels is corrupted, does selecting top-confidence/variability samples still help or hurt?

4.How sensitive are results to the ECE binning strategy (number of bins)? Please report calibration reliability diagrams and standard deviations across runs.

5.Could you comment on potential negative interactions between the precomputation model and the final model (same architecture/initialization) — i.e., is there a risk of overfitting via the selection process?

6.Any plans to extend the idea to non-classification settings (e.g., retrieval, ranking) where “gold label probability” is not directly available?

---

### Meta-Review · Area_Chair_Hu1n · 2026-01-04

**Summary:**

This paper presents a supervised contrastive learning method which uses data maps to guide sample selection. Reviewers agree that the paper is clearly written and the method is intuitively motivated. However, there are common concerns that the contribution is incremental and that the empirical results are not strong enough to support the paper’s central claims. Key concerns include limited novelty beyond prior work on hard positives/negatives and data maps, relatively weak or potentially unfair baselines, heuristic and sensitive hyperparameter choices, and the added complexity and potential bias of the two-stage training pipeline. Specifically, Reviewer ycnJ raises serious concerns that the experiments do not convincingly demonstrate either the sufficiency or necessity of the proposed method. The authors didn't submit a rebuttal to address these concerns. Given these, I recommend rejecting the paper in its current form.

**Reviewer Concerns:**

The authors didn't submit a rebuttal to address these concerns.

**Reviewer Scores:**

It is unlikely that reviewers would change their scores, as no rebuttal was submitted.

---

### Decision · Program_Chairs · 2026-01-26

Reject